# Inducing Meaningful Units from Character Sequences with Dynamic Capacity Slot Attention

**Melika Behjati**  *melika.behjati@epfl.ch*
*École Polytechnique Fédérale de Lausanne (EPFL)*
*Idiap Research Institute*

**James Henderson**  *james.henderson@idiap.ch*
*Idiap Research Institute*

**Reviewed on OpenReview:** *https: // openreview. net/ forum? id=m8U9rSs6gU*

## Abstract

Characters do not convey meaning, but sequences of characters do. We propose an unsupervised distributional method to learn the abstract meaningful units in a sequence of characters. Rather than segmenting the sequence, our Dynamic Capacity Slot Attention model discovers continuous representations of the *objects* in the sequence, extending an architecture for object discovery in images. We train our model on different languages and evaluate the quality of the obtained representations with forward and reverse probing classifiers. These experiments show that our model succeeds in discovering units which are similar to those proposed previously in form, content and level of abstraction, and which show promise for capturing meaningful information at a higher level of abstraction.

## 1 Introduction

When we look at a complex scene, we perceive its constituent objects, and their properties such as shape and material. Similarly, what we perceive when we read a piece of text builds on the word-like units it is composed of, namely *morphemes*, the smallest meaning-bearing units in a language. This paper investigates deep learning models which discover abstract representations of such meaningful units from the distribution of character sequences in natural text.

In recent years, there has been an emerging interest in unsupervised object discovery in vision (Eslami et al., 2016; Greff et al., 2019; Engelcke et al., 2020; Elsayed et al., 2022; Seitzer et al., 2023). The goal is to segment the scene into its objects without supervision and obtain an object-centric representation of the scene. These representations should lead to better generalization to unknown scenes, and additionally should facilitate abstract reasoning over the image. Locatello et al. (2020) proposed a relatively simple and generic algorithm for discovering objects called Slot Attention, which iteratively finds a set of feature vectors (i.e., slots) which can bind to any object in the image through a form of attention.

Inspired by this line of work in vision, our task is to learn a set of abstract continuous representations of the *objects* in text. We adapt the Slot Attention module (Locatello et al., 2020) for this purpose, extending it for discovering the meaningful units in natural language character sequences. This makes our proposed task closely related to unsupervised morphology learning (Creutz, 2003; Narasimhan et al., 2015; Eskander et al., 2020). However, there are fundamental differences between our task and morphology learning. First, we learn to represent a text with a set of vectors, which are not explicitly tied to the text segments. Second, our model learns its representations by considering the entire input sentence, rather than individual space-delimited words. These properties of our induced representations make our method appropriate for inducing meaningful units as part of deep learning models.

In particular, we integrate our unit discovery method on top of the encoder in a Transformer auto-encoder (Vaswani et al., 2017), as depicted in Figure 1, and train it with an unsupervised sentence reconstruction

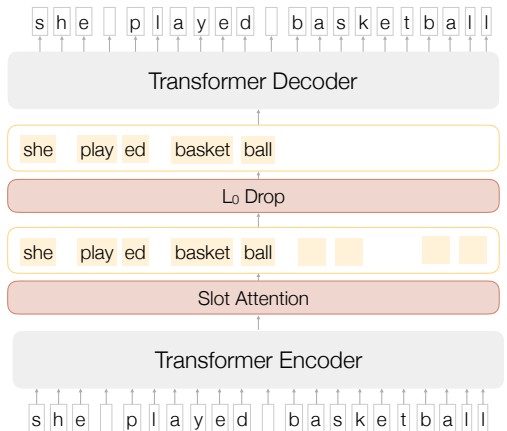

Figure 1: Dynamic Capacity Slot Attention. First, the Transformer encoder encodes the sequence and then, Slot Attention computes the slot vectors (highlighted text). Next, the $L_0$Drop layer dynamically prunes out the unnecessary slots and the decoder finally reconstructs the original sequence.

objective. This setting differs from previous work on Slot Attention, which has been tested on synthetic image data with a limited number of objects (Locatello et al., 2020). We propose several extentions to Slot Attention for the domain of real text data. We increase the capacity of the model to learn to distinguish a large number of textual units, and add the ability to learn how many units are needed to encode sequences with varying length and complexity. Thus, we refer to our method as Dynamic Capacity Slot Attention. Additionally, as a hand-coded alternative, we propose stride-based models and compare them empirically to our slot attention based models.

To evaluate the induced representations we both qualitatively inspect the model itself and quantitatively compare it to previously proposed representations. Visualisation of its attention patterns shows that the model has learned representations similar to the contiguous segmentations of traditional tokenization approaches. Trained probing classifiers show that the induced units capture similar abstractions to the units previously proposed in morphological annotations (MorphoLex (Sánchez-Gutiérrez et al., 2018; Mailhot et al., 2020)) and tokenization methods (Morfessor (Virpioja et al., 2013), BPE (Sennrich et al., 2016)). To compare two representations, we propose to do this probing evaluation in both directions, to compare both informativeness and abstractness of the units. These evaluations show promising results in the ability of our models to discover units which capture meaningful information at a higher level of abstraction than characters.

To summarize, our contributions are as follows: (i) We propose a novel model for learning meaningful units from a sequence of characters (Section 2). (ii) We propose simple stride-based models which serve as strong baselines for evaluating such unsupervised models (Section 2.3). (iii) We analyze the induced units by visualizing the attention maps of the decoder over the slots and observe the desired sparse and contiguous patterns (Section 4.3). (iv) We propose a bi-directional probing method to evaluate such models by comparing the induced units to given units in terms of both their informativeness and their abstractness (Section 2.4). (v) We show that our induced units capture meaningful information at an appropriate level of abstraction by probing their equivalence to previously proposed meaningful units (Section 4.4). (vi) We evaluate the induced units in a downstream task to show its potential benefits over character-level and segmentation-based representations (Section 4.5).

## 2 Approach

The unsupervised induction of abstract meaningful units from sequences of characters is a novel task which requires both novel methods and novel evaluations. It should allow the induction of meaningful units as part of deep learning models. In this section we define the task, propose two types of models, and propose a probing-based evaluation.

---

**Algorithm 1** Slot Attention module (Locatello et al., 2020). $q, k, v$ map the slots and inputs to a common dimension $D$ and $T$ denotes the number of iterations.

---

**Require:** inputs $\in \mathrm{R}^{N \times D_{input}}$, slots $\sim \mathcal{N}(\mu, diag(\sigma)) \in \mathrm{R}^{K \times D_{slots}}$

  inputs = LayerNorm(inputs)
  **for** i = 1 **to** T **do**
    slots_prev = slots
    slots = LayerNorm(slots)
    attn = Softmax($\frac{1}{\sqrt{D}}$ $k$(inputs).$q$(slots)$^T$, axis = 'slots')
    updates = WeightedMean(weights = attn + $\delta$ , values = $v$(inputs))
    slots = GRU(states = slots_prev, input = updates )
    slots += MLP(LayerNorm(slots))
  **end for**
  **return** slots

---

## 2.1 Problem Formulation

Given a sequence of $N$ characters $X = x_1 x_2 \ldots x_N$, we want to compute a set of vectors $M = \{m_1, \ldots, m_K\}$ which best represents $X$ at a higher level of abstraction. We expect that each vector (i.e. slot) in the set $M$ should represent a meaning-bearing unit in the text $X$, so that the induced representation $M$ captures the desired level of abstraction. As an example, consider the sequence *"she played basketball"*, where we expect each of our vectors to represent one of the morphemes of the sequence, namely {*she, play, -ed, basket, -ball*}. Although the meaning-bearing units are often assumed to come from a fixed vocabulary, we do not assume this and compute the vectors dynamically from the full text.

## 2.2 Dynamic Capacity Slot Attention

We learn our representations through encoding the input sequence into slots and then reconstructing the original sequence from them. In particular, we use an auto-encoder structure where slots act as the bottleneck between the encoder and the decoder. Figure 1 shows an overview of our proposed model, Dynamic Capacity Slot Attention. First, we encode the input character sequence by a Transformer encoder (Vaswani et al., 2017), which gives us one vector per character. Then, we apply our higher-capacity version of a Slot Attention module (Locatello et al., 2020) over the encoded sequence, to learn the slots. Intuitively, Slot Attention will learn a soft clustering over the input where each cluster (or respectively slot) corresponds to a candidate meaningful unit in the sequence. To select which candidates are needed to represent the input, we integrate an $L_0$ regularizing layer, i.e., $L_0$Drop layer (Zhang et al., 2021), on top of the slots. Since the maximum number of slots is fixed during the course of training, this layer ensures that the model only uses as many slots as necessary for the particular input. This stops the model from converging to trivial solutions for short inputs, such as passing every character through a separate slot. Finally, the Transformer decoder reconstructs the input sequence autoregressively using attention over the set of slots.

**Encoder.** We use the Transformer encoder architecture for encoding our sequence (Vaswani et al., 2017) and obtain the representation $X' = x'_1 x'_2 \ldots x'_N$ from our input sequence $X$.

**Slot Attention for Text.** After encoding the character sequence, we use our extended version of Slot Attention for discovering meaningful units of the input character sequence. Slot Attention is a recent method for unsupervised object representation learning in vision (Locatello et al., 2020). It learns a set of feature vectors (slots) by using an iterative attention based algorithm. Algorithm 1 shows the pseudo code of this method. Abstractly, in every iteration, it takes the following steps. First, it computes an attention map between the slots and the inputs, with slots acting as queries and inputs as keys. Then, it normalizes the attention map over the slots, which makes the slots compete for representing each token of the input. Afterwards, it computes the slots' updates as the (normalized) weighted mean over the attention weights and the input values. Finally, it updates the slots through a Gated Recurrent Unit (GRU) (Cho et al., 2014) followed by a residual MLP. This process iterates a fixed number of times.

In Locatello et al. (2020), the slots are initialized randomly by sampling from a Normal distribution with shared learnable parameters $\mu$ and $\sigma$, i.e.,

$$\text{slot}_i \sim \mathcal{N}(\mu_{\text{shared}}, \sigma_{\text{shared}}). \tag{1}$$

In other words, the initial value of slots are independent samples of a single Normal distribution with learnable parameters. We found in our experiments that this initialization method does not lead to good results in the text domain (see Appendix A.1 for a comparison). In contrast with the experimental setting of Locatello et al. (2020), which used artificially generated data with a small number and range of objects, real language requires learning about a very large vocabulary of morphemes, and each example can have a large number of morphemes. This suggests that a model for language needs a more informative initialization with more trainable parameters, in order to have the capacity to learn about this large vocabulary and distinguish all the objects in a long sentence.

To investigate this issue, we propose another initialization for adapting Slot Attention to text. We consider a separate learnable $\mu$ per slot and we fix the $\sigma$ to a predefined value for all the slots. Namely, the slots are initialized as

$$\text{slot}_i \sim \mathcal{N}(\mu_i, \sigma_{\text{constant}}). \tag{2}$$

By assigning a separate $\mu$ for each slot, the initialization has many more trainable parameters. This allows the model to learn about different kinds of units, such as ones that occur at different positions, or ones that have different types of forms, but we do not make any assumptions about what those differences might be. In addition, the intuition behind fixing the $\sigma$ is to prevent it from collapsing to zero. In particular, since the number of possible n-grams in text is finite but the slots can have any continuous value in the space of $\mathbb{R}^{D_{slots}}$, the slots tend to learn an arbitrary mapping from n-grams in the input to the slots while turning $\sigma$ to zero. In this case, there is no need for the slots to learn the underlying meaning-bearing units. By fixing the $\sigma$ to a nonzero value, the initialization effectively introduces noise into slot vectors and thereby limits the amount of information which can be passed through each slot, from the information theoretic point of view. This bottleneck forces the slots to compress information in a meaningful way. We then obtain the set of slots $M = \{m_1 \dots m_K\} = \text{SlotAttention}(X')$.

**Neural Sparsification Layer: $L_0$Drop.** The number of units needed to represent a sequence varies among different sequences in the data. In contrast to the object discovery work where the data is generated synthetically and thereby, the number of objects in the scene is known beforehand, we do not make any assumptions about the number of units required. Therefore, we consider an upper-bound over the number of required units and prune the extra ones per input sequence. We accomplish this goal by using a neural sparsification layer called $L_0$Drop (Zhang et al., 2021). It allows our model to dynamically decide on the number of required units for every input sequence.

This layer consists of stochastic binary-like gates $\boldsymbol{g} = g_1 \dots g_K$ that for every input $m_i$ works as

$$L_0\text{Drop}(m_i) = g_i m_i. \tag{3}$$

When $g_i$ is zero the gate is closed and when it is one the whole input is passed. Each gate is a continuous random variable in the $[0, 1]$ interval, sampled from a hard-concrete distribution (Louizos et al., 2018). This distribution assigns most of its probability mass over its endpoints (i.e., 0 and 1) in favour of the sparsification goal. Specifically, $g_i \sim \text{HardConcrete}(\alpha_i, \beta, \epsilon)$ where $\beta$ and $\epsilon$ are hyperparameters. $\alpha_i$ is predicted as a function of the encoder output $m_i$, i.e., $\log \alpha_i = m_i w^T$, where $w$ is a learnable vector. This allows the model to dynamically decide which inputs to pass and which ones to prune. The $\mathcal{L}_0$ penalty, which yields the expected number of open gates, is computed as:

$$\mathcal{L}_0(M) = \sum_{i=1}^{k} 1 - p(g_i = 0 | \alpha_i, \beta, \epsilon), \tag{4}$$

where the probability of $g_i$ being exactly 0 is provided in closed form in Louizos et al. (2018). We follow the same approach as Louizos et al. (2018) at evaluation time and consider the expectation of each gate as its value. We refer to the pruned slots after applying the $L_0$Drop layer as $M' = m'_1 \dots m'_K$.

**Decoder.** Lastly, we regenerate the input sequence from the set of slots by using a simple, shallow decoder. To this end, we use a one-layer Transformer decoder (Vaswani et al., 2017) with a single attention head over the slots. A simple decoder forces the slots to learn representations with a straightforward relationship to the input, which we expect to be more meaningful. In other words, we do not use a powerful decoder because it would be able to decode even low-quality representations of the input, which are less meaningful (Bowman et al., 2015).

**Training Objective.** We train our model end-to-end by using Gumble trick for sampling HardConcrete variables (Maddison et al., 2017; Jang et al., 2017). The training objective is

$$
\begin{aligned}
\mathcal{L}_{\text{rec}}(X, M') + \lambda\, \mathcal{L}_0(M) \quad &= \quad -\log\left(\mathrm{E}_{\boldsymbol{g}}[p(X|M')]\right) + \lambda\, \mathcal{L}_0(M) \\
&\leq \quad \mathrm{E}_{\boldsymbol{g}}[-\log p(X|M')] + \lambda\, \mathcal{L}_0(M) \quad = \quad \mathcal{L}(X),
\end{aligned}
\tag{5}
$$

which consists of the reconstruction loss from the decoder ($L_{\text{rec}}$) and the $L_0$ penalty for the open gates. Hyperparameter $\lambda$, the sparsification level, controls the trade-off between the two losses. In practice, we find that in order to impose enough sparsity in the slots, we should slightly increase $\lambda$ during the course of training using scheduling techniques.

**Training Objective with Targeted Sparsity.** Controlling the level of sparsity by setting $\lambda$ can be difficult, so we provide an alternative version of the loss with an explicit target sparsity rate $r$ (Mai & Henderson, 2022). We replace the $L_0$ term in Equation 5 with $\max(L_0, 1/r \times N)$, to have

$$
\mathcal{L}(X) = \mathrm{E}_{\boldsymbol{g}}[-\log p(X|M')] + \lambda \max(\mathcal{L}_0(M), 1/r \times N)
\tag{6}
$$

where $1/r$ indicates the proportion of slots that we want to keep open and $N$ is the maximum input length. In this setup, the model will stop closing more gates when $L_0$ reaches the target $1/r \times N$ [1].

## 2.3 Stride-based Models

We propose a simple hand-crafted alternative model to our induced units which can gain acceptable results in terms of performance. We design this model by replacing our Dynamic Capacity Slot Attention module with a linear-size bottleneck. In particular, we take 1 out of every $k$ encoder outputs and down project them ($\mathrm{R}^{D_{input}} \to \mathrm{R}^{D_{slots}}$) to obtain the representations. In other words, we only pass certain encoder outputs based on their position and drop the rest.

$$
M = \mathrm{DownProject}(x'_1 x'_{k+1} x'_{2k+1} \ldots x'_{nk+1})
$$

where $n = \lfloor \frac{N-1}{k} \rfloor$. We can get different alternative models by varying the stride $k$. The training objective is the reconstruction loss $\mathcal{L}_{\text{rec}}(X, M) = -\log p(X|M)$. The idea of using stride-based models has also been used in Clark et al. (2022); Tay et al. (2022) in a different context and setup.

## 2.4 Bi-directional Probing Evaluation

Since the task is unsupervised and we do not evaluate using artificially generated data, there is no obvious gold standard for what units a method should learn. To quantitatively evaluate how well a model performs, we freeze the trained model and train probing classifiers to measure to what extent the discovered units capture the same information as previously proposed meaningful units. We use multiple previously proposed representations which have been shown to be good levels of abstraction for a range of NLP applications. If the induced representation separates the required information into units with a similar level of abstraction, then we can expect that they will also be effective in NLP applications.

We propose to measure whether two representations provide the same level of abstraction by measuring whether there is a one-to-one mapping between their respective units such that two aligned units contain the same information, meaning that each unit can be predicted from the other. Predicting the previously proposed unit from our induced unit is a probing task, as used in previous work (Belinkov & Glass, 2019),

---

[1]Note that this term would be effective when $1/r \times N \leq K$, where $K$ is the maximum number of slots.

which we specify as *forward probing.* We refer to the prediction of our induced unit from the previously proposed unit as a *reverse probing* task, which we believe is a novel form of evaluation[2]. We compare various models on their trade-off between forward and reverse probing.

**Forward probing.** This is the common way of probing where we want to measure if our induced units include the information about the target representation. We train a classifier with shared parameters on each of our slots individually and obtain a *set* of predictions $\{f(m'_1), f(m'_2), \ldots, f(m'_K)\}$, one per slot. As we are dealing with a set, we need to find a one-to-one matching between the classifier's predictions and the target tokens. Therefore, we follow Locatello et al. (2020) to use the Hungarian matching algorithm (Kuhn, 1955) for finding the match which minimizes the classification loss. The same alignment method is applied both at training and at testing time.

We consider the complete set of slots after applying the $L_0$Drop layer as the inputs to our classifier. Slots whose $L_0$Drop gate is closed are simply input as zero vectors. This gives us a fixed number of vectors. The two sides of matching should have the same size to obtain a one-to-one match, therefore, we add an extra target label (i.e., *empty*) for representing the pruned slots. Due to the fact that many slots are pruned out, considering a measure like accuracy could be misleading, since a classifier which outputs *empty* label will achieve very high accuracy. Therefore, we build a confusion matrix as follows. We consider all *non-empty* labels as positive and the *empty* ones as negative, and we report precision (P), recall (R) and F1 measure, to better reflect what the slots have learned.

**Reverse probing.** To evaluate whether the induced units capture the same level of abstraction, we also need to evaluate whether the induced units abstract away from the same information as the previously proposed units. We propose to measure this by training reverse probing classifiers, which predict each induced unit from its aligned target unit. Because the induced unit is a continuous vector, we predict the parameters of a $d$-dimensional Gaussian distribution with diagonal covariance, and measure the density function of the induced vector in this distribution. The probe is trained to minimise this negative-log-density function, and this loss on the test set is used as our evaluation measure.[3]

## 3 Related Work

**Unsupervised Object Discovery.** There is a recent line of research in the image domain for discovering objects in a scene without explicit supervision, and building an object-centric representation of them. Most of this work is built around the idea of compositionality of the scenes. MONet (Burgess et al., 2019) and GENESIS (Engelcke et al., 2020) similarly use a recurrent attention network for learning the location masks of the objects. Greff et al. (2016; 2017; 2019); Van Steenkiste et al. (2018); Emami et al. (2021) model the scene as a spatial Gaussian mixture model. Furthermore, AIR network (Eslami et al., 2016) and its variants (Crawford & Pineau, 2019; Lin et al., 2020) model objects from a geometric perspective by defining three specific latent variables. Lately, Locatello et al. (2020) propose an attention-based algorithm (namely Slot Attention) to learn object representations (slots) which have been followed by Singh et al. (2022); Sajjadi et al. (2022); Seitzer et al. (2023); Singh et al. (2023) and Kipf et al. (2022); Elsayed et al. (2022) further expanded it to the video domain. In contrast to this line of work in vision, our approach is specifically designed for text. We use additional components (e.g., $L_0$Drop layer) in our architecture to resolve the requirements of modeling textual data. Furthermore, our model is trained and evaluated on real text datasets, in contrast to these previous models which have only been shown to be effective on synthetic scenes.

**Unsupervised morphology learning.** This subject has been of interest for many years in the NLP field (Elman, 1990; Creutz & Lagus, 2002; Baroni et al., 2002). Morphemes have strong linguistic motivations, and are practically important in many downstream tasks because they are the smallest meaning-bearing units in a language (Can & Manandhar, 2014). Many approaches have been proposed for discovering the underlying morphemes or morpheme segmentations. Morfessor variants are based on probabilistic machine learning methods (MDL, ML, MAP) for morphological segmentation (Creutz, 2003; Creutz & Lagus, 2002; 2005; 2007; Virpioja et al., 2013). Some researchers take a Bayesian approach for modeling word formation (Poon et al.,

---

[2]See Figure 8 in the Appendix for an illustration of forward and reverse probing

[3]Note that as we are modelling a continuous distribution, the density function can have values higher than 1 and therefore, the negative-logarithm of it could be negative.

2009; Narasimhan et al., 2015; Bergmanis & Goldwater, 2017; Luo et al., 2017). Adaptor Grammars are another approach for modeling morphological inflections (Sirts & Goldwater, 2013; Eskander et al., 2016; 2019; 2020). In addition, Xu et al. (2018; 2020) built their models upon the notion of paradigms, set of morphological categories that can be applied to a set of words. Moreover, Soricut & Och (2015); Üstün & Can (2016) extract morphemes by considering the semantic relations between words in the continuous embedding space. Cao & Rei (2016) propose to learn word embeddings by applying a bi-directional RNN with attention over the character sequence. Furthermore, Ataman et al. (2020) model word formation as latent variables which mimic morphological inflections in the task of machine translation.

Our work differs from the previous work in classical morphology learning in two ways. First, instead of explicitly discovering morphemes, we learn a set of continuous vector representations of the input, which would then need to be processed to extract the morphemes. The model itself has no explicit relation between these unit representations and segments of the input. Second, our model learns representations of an entire input sentence, rather than individual space-delimited words. This makes fewer assumptions about morphemes, and considers the context of the words in a sentence. Our work is similar to Ataman et al. (2020) in modeling morphology implicitly in the latent space. However, we employ a self-supervised objective for our purpose which is more general compared to their supervised loss, as we do not need labeled data.

**Unsupervised character segmentation.** Learning to segment a character sequence in unsupervised fashion is another relevant area to our work. Chung et al. (2017) propose Hierarchical Multi-scale RNNs for modeling different levels of abstractions in the input sequence. In the language modeling task, they observe that the first layer is roughly segmenting the sequence into words, namely at space boundaries. Sun & Deng (2018) propose Segmental Language Models for Chinese word segmentation. Moreover, in (Kawakami et al., 2019), the authors design a model to learn the latent word segments in a whitespace-removed character sequence with a language modeling objective. As we mentioned earlier, we learn continuous vector representations of text which is different from explicitly detecting discrete character segments.

**Learning intermediate representations from characters.** Recently, a line of work has been proposed to model language at the level of characters and then aggregate the characters into subword-like units and have the core Transformer layers on top of them. CANINE Clark et al. (2022) learns these units by applying local attention and strided convolutions and proves the usefulness of their model in multilingual tasks. Charformer Tay et al. (2022) learns a representation for each character by computing a weighted sum over character k-grams that this character is involved in and then performs mean pooling to downsample the representation size. The goal of our works differs from theirs as we are focusing on learning meaning-bearing units at the level of morphemes and we are not focusing on improving downstream tasks' performance directly.

**Unsupervised speech unit discovery.** Unsupervised unit discovery has also been studied in the speech domain, where the speech data includes both acoustic and language information. Wav2vec 2.0 Baevski et al. (2020) encodes the raw input using a convolutional neural network. Then, it learns units in the representation by discretizing the output of the feature encoder using product quantization Jegou et al. (2010). HuBERT Hsu et al. (2021) follows the same architecture as Wav2vec 2.0 but they discover acoustic units leveraging K-means algorithm. Authors in Tjandra et al. (2020) extract subword units given speech audio by a variational autoencoder with a finite set of discrete latent variables (aka limited codebook) called vector quantized variational autoencoder (VQ-VAE) Van Den Oord et al. (2017).

**Subword discovery algorithms.** This set of algorithms have become a standard component of NLP models in recent years. Byte-Pair-Encoding (BPE) (Sennrich et al., 2016) iteratively merges the two consecutive tokens with the highest frequency. Word-piece (Schuster & Nakajima, 2012), sentence-piece (Kudo & Richardson, 2018) and unigram LM (Kudo, 2018) are other similar subword tokenization algorithms. In contrast to these methods, which mostly use local statistical information of the data, our model is trained over complete sentences to learn an abstract sophisticated representation.

## 4  Experiments

We train our models as auto-encoders and evaluate the resulting induced units. In addition to considering the ability of the induced representations to reconstruct the input (their supervised objective), we evaluate

Table 1: Reconstruction error on training and test set among different languages.

| language | slot attention(r=6) | | stride=6 | |
|---|---|---|---|---|
| | train | test | train | test |
| DE | **0.0022** | **0.0039** | 0.0174 | 0.0213 |
| EN | **0.0014** | **0.0033** | 0.0120 | 0.0154 |
| CS | **0.0021** | **0.0035** | 0.0089 | 0.0137 |
| FI | **0.0028** | **0.0045** | 0.0097 | 0.0135 |
| FR | **0.0043** | **0.0075** | 0.0182 | 0.0229 |
| ES | **0.0014** | **0.0036** | 0.010 | 0.0149 |

our unsupervised model both by visualizing attention maps to show what characters the slots correspond to (Section 4.3), and by bi-directional probing of the slot vectors (Section 4.4). In addition, we show the potential usefulness of our induced slots in a downstream task (Section 4.5).

## 4.1 Experimental Setup

We apply our model to languages from different morphological typologies. We select English (EN), German (DE), French (FR), Spanish (ES) and Czech (CS) from the fusional family and Finnish (FI) from the agglutinative typology. For English we use the raw Wikitext2 dataset (Merity et al., 2017). For the rest, we use Multilingual Wikipedia Corpus (MWC) (Kawakami et al., 2017).

As for the models, we use a standard Transformer architecture (Vaswani et al., 2017) with model dimension 256. The encoder consists of 2 layers with 4 self-attention heads and the decoder consists of 1 layer with 1 self-attention head and 1 attention head over the slots. We feed in the sentences with less than 128 characters to our model and consider the maximum number of slots as 64 (half of the maximum input length). In addition, we take the dimension of slots as 128. We scheduled the $\lambda$ parameter in the training loss to start with a low value and exponentially increase it every 10 epochs until it reaches a certain limit. We obtain this limit manually in a way that the final number of open gates roughly equals the average number of BPE tokens in a sequence. More details of the settings are available in the Appendix B.

For the forward probing classifier, we train a 2 layer MLP as our probing classifier $f$, which predicts the target token given the slot vector. For the reverse probing classifier, we first pass the target tokens into an Embedding layer and then apply a 2 layer MLP on top of each embedding individually to predict the parameters of a Gaussian distribution over slot vectors.

## 4.2 Reconstruction Results

The models are trained to reconstruct the input character sequence, so the reconstruction error (reported in Table 1) indicates how effectively the model can compress information about the text in the available units' vectors. We compare the slot attention based model with targeted sparsity objective (Equation (6)), with the stride-based models (stride=6) to ensure that both models have the same average number of units. In all languages, slot attention based models are better able to reconstruct the sequence. This indicates that the slot attention vectors are better than the stride-based vectors at capturing information about the structure and meaning of the input, thereby providing better signals to the decoder to reconstruct the sequence.

## 4.3 Visualization

In order to show some qualitative results of our model, we visualize the attention maps for generating every output, shown in Figure 2. In particular, we show the attention of the decoder over slots when generating every output character [4]. Interestingly, although we do not impose any sparsity in the decoder's attention weights, the attention maps are quite sparse. Namely, at each generation step only a few slots are attended, and each slot is attended while generating only a few characters. In addition, although we do not impose

---

[4]We observe the same pattern in the attention of slots over the input and have illustrated some examples in Appendix A.2.1

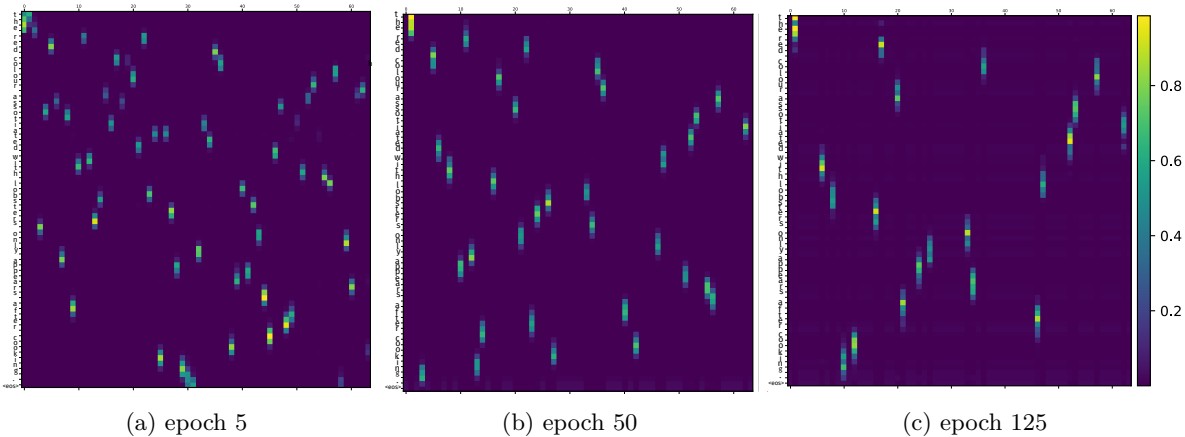

Figure 2: Attention of the decoder over slots (x-axis) for generating every target character (y-axis) after different training epochs, for target sequence "*the red colour associated with lobsters only appears after cooking.*".

any bias towards discovering segments of the input, the characters which are generated while attending to a given slot are contiguous in the string (the vertical bands in Figure 2). We believe that the emergence of contiguous spans is a result of the bottleneck we create with our Dynamic Capacity Slot Attention. This means that the model is trying to put correlated information about the input in the same vector, so that it can represent the string more efficiently. The strongest correlations in the character string are local in the input, so each slot tends to represent characters which are next to each other in the input.

In early steps of training, when the sparsity ratio ($\lambda$) is small, each slot tends to represent a bigram of characters (2a) and later on, trigrams (2b). These observations confirm the necessity of the $L_0$Drop layer for converging to better units. In particular, as the ratio increases, the number of active slots reduces and they become more specialized in representing contiguous meaning-bearing units of input. For instance, the word *cooking* in 2c is represented by two slots *cook* and *ing*. That these segments roughly correspond to the morphemes which we want the model to discover, is verified quantitatively in the probing experiments in Section 4.4. We provide attention map visualizations for other languages and stride-based models in Appendix A.2.

### 4.3.1 Learned Positional Information

We observed similar attention patterns for the slots across different input samples and thereby, we conjecture that there must be a correlation between what the slots have learned and the corresponding positions in the input sequence. To evaluate this, we average the attention maps between all the samples from the test set and visualize them in Figure 3. The averaged attention map verifies our hypothesis that the slots are highly correlated with the position of characters in the sequence. We think that this average is related to what the $\mu_i$ is learning, but that after the Slot Attention iterations the slots carry more information than just positions.

### 4.4 Probing Results

Given a target set of units and a trained model for finding units, we align the two sets of units, and use forward probing to see whether each induced unit contains as much information as its aligned target unit and use reverse probing to see whether each induced unit contains more information than its aligned target unit. Optimizing both measures requires having the same information as each target unit in its associated induced vector, and hence having the same level of abstraction.

The primary mechanism we have to control the amount of information in an induced representation is to control the number of units. More units means that there are more opportunities for the alignment to find a

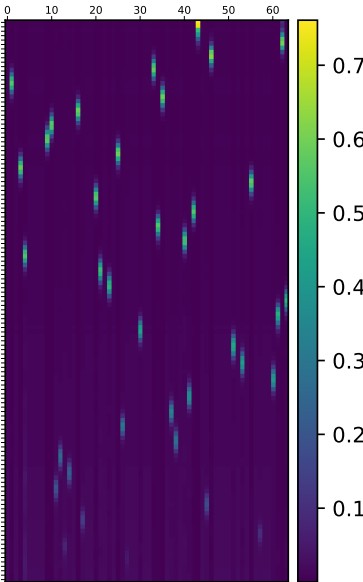

Figure 3: Average attention maps across all the test set samples. x-axis corresponds to slots and y-axis is the decoder's output position. The later positions (bottom) occur in fewer examples, but almost all slots show a clear preference for specific positions.

unit which correctly predicts each target unit, but makes it harder to predict all the units from the available target units. We take two approaches to controlling the number of induced units. First, we fix the number of units to a target number, which allows an efficient comparison of models. Second, we vary the number of units, which allows us to evaluate how well different models can match the informativeness trade-off of the target representation.

**Probing data.** We consider three target representations which either have linguistic or practical evidence of being effective in NLP applications. For the languages for which an in-depth linguistic morphological analysis is available, we compare to these morphemes (i.e., EN and FR) (Žabokrtský et al., 2022; Sánchez-Gutiérrez et al., 2018; Mailhot et al., 2020). Alternatively, as linguistically inspired units, we have the Morfessor (Virpioja et al., 2013) outputs, and BPEs (Sennrich et al., 2016) as frequency-based subwords.

**Targeted informativeness.** As a less computationally expensive initial evaluation, we use hyperparameters to set the number of induced units to approximately the same as the number of target BPE tokens. For the stride-based models, we simply set the stride according to the average target number of tokens (i.e., stride= 6 or = 5). For the slot attention models, we set the $r$ in Equation (6) to be equal to the stride value.

We use probing to compare slot attention and stride-based models to an uninformative baseline representation (untrained), thereby controlling for the power of the probing classifier to learn arbitrary mappings (Conneau et al., 2018; Oord et al., 2018). This baseline is the set of slot vectors output by a randomly-initialized slot attention model without any training.

Table 2 shows the results of the forward and reverse probing tasks on different languages. As the results show, the trained slots achieve much higher performance in all the tasks in comparison to the random baselines. Our model achieves very high precision in predicting the *non-empty* labels. Its performance is weaker on the recall side, but here the improvement over the untrained model is even more pronounced. This seems to be due to the imbalance between *empty* and *non-empty* labels in the training set, where the *empty* labels comprise around 66% of the data for the probing classifier. For this reason, below we will use F1 as our overall performance measure for forward probing.

For most languages and targets, our slot attention model performs better than the comparable stride-based model. In the cases where the stride-based model has a higher F1, its worse score for reverse probing suggest

Table 2: Forward and reverse probing results on different languages, with a target number of units. Note that human-annotated morphemes are only available for En and FR.

| lang | Model | BPE P | R | F1 | Rev | Morfessor P | R | F1 | Rev | Morphemes P | R | F1 | Rev |
|------|-------|-------|---|----|-----|-------------|---|----|-----|-------------|---|----|-----|
| EN | untrained | 0.74 | 0.10 | 0.17 | $-47.12$ | 0.71 | 0.11 | 0.19 | $-36.53$ | 0.69 | 0.11 | 0.19 | $-50.29$ |
|    | stride$= 6$ | 0.95 | 0.66 | 0.76 | $-105.1$ | 0.94 | 0.66 | **0.76** | $-107.9$ | 0.91 | 0.64 | 0.74 | $-116$ |
|    | slot-attn$(r = 6)$ | 0.96 | 0.68 | **0.78** | **-150.1** | 0.93 | 0.63 | 0.74 | **-140** | 0.91 | 0.70 | **0.78** | **-123.7** |
| FI | untrained | 0.74 | 0.09 | 0.17 | $-72.03$ | 0.90 | 0.06 | 0.11 | $-39.23$ | - | - | - | - |
|    | stride$= 5$ | 0.94 | 0.63 | **0.74** | **-123.7** | 0.89 | 0.43 | 0.57 | **-112.6** | - | - | - | - |
|    | slot-attn$(r = 5)$ | 0.94 | 0.58 | 0.71 | $-104.7$ | 0.89 | 0.50 | **0.63** | $-80.62$ | - | - | - | - |
| FR | untrained | 0.75 | 0.08 | 0.14 | $-51.59$ | 0.72 | 0.11 | 0.19 | $-28.05$ | 0.74 | 0.11 | 0.19 | $-68$ |
|    | stride$= 5$ | 0.94 | 0.69 | 0.79 | $-129$ | 0.93 | 0.65 | 0.76 | $-120.3$ | 0.91 | 0.68 | 0.77 | **-204.3** |
|    | slot-attn$(r = 5)$ | 0.96 | 0.70 | **0.80** | **-164.2** | 0.94 | 0.68 | **0.78** | **-160.8** | 0.91 | 0.72 | **0.80** | $-190.2$ |
| ES | untrained | 0.71 | 0.09 | 0.17 | $-37.85$ | 0.71 | 0.11 | 0.19 | $-44.63$ | - | - | - | - |
|    | stride$= 5$ | 0.94 | 0.67 | 0.77 | **-103.5** | 0.93 | 0.65 | 0.75 | **-131.6** | - | - | - | - |
|    | slot-attn$(r = 5)$ | 0.94 | 0.69 | **0.78** | $-94.83$ | 0.93 | 0.66 | **0.76** | $-125.7$ | - | - | - | - |
| DE | untrained | 0.91 | 0.08 | 0.16 | $-46.91$ | 0.76 | 0.08 | 0.15 | $-64.87$ | - | - | - | - |
|    | stride$= 5$ | 0.95 | 0.66 | 0.77 | $-170.1$ | 0.91 | 0.58 | 0.69 | $-156.1$ | - | - | - | - |
|    | slot-attn$(r = 5)$ | 0.97 | 0.68 | **0.79** | **-190.8** | 0.93 | 0.66 | **0.76** | **-157.9** | - | - | - | - |
| CS | untrained | 0.94 | 0.08 | 0.16 | $-22.98$ | 0.86 | 0.05 | 0.10 | $-34.69$ | - | - | - | - |
|    | stride$= 5$ | 0.94 | 0.63 | **0.74** | $-104.5$ | 0.90 | 0.52 | **0.65** | $-87.16$ | - | - | - | - |
|    | slot-attn$(r = 5)$ | 0.96 | 0.62 | **0.74** | **-164.5** | 0.95 | 0.47 | 0.62 | **-148.1** | - | - | - | - |

that this may just be the result of an informativeness trade-off. For the majority of cases where the slot attention model has a higher F1, its reverse probing score is also better. The performance gain of our slot attention models over the stride-based models is particularly noticeable for the two languages where we have real morpheme annotations, which implies that our slot attention based method is more effective in discovering linguistically meaningful units. The agglutinative language (Finnish) is generally harder than the fusional languages, but slot attention does relatively well at capturing the linguistically-motivated Morfessor targets compared to the stride-based model. We further show some examples from the predictions of the forward probing classifiers in the Appendix A.3.

**Informativeness trade-off.** The advantage of our two-directional probing evaluation is that we can directly compare the informativeness trade-off of different models by plotting their probing results in two-dimensional graphs. To vary this trade-off, we train stride-based models with different strides, stride$=\{2, 3, 4, 5, 6\}$, and slot attention based models with different $r$, where $r =$ stride, for the same strides. This leads to the two models having the same range of values for their number of valid units.

Figure 4 plots this informativeness trade-off for the same languages and target representations as in Table 2. Better results are nearer to the lower right corner of the plot. Overall, the slot attention based models provide better representations than the stride-based models. About half the plots show a clear trend in favour of slot attention based models, and none of the plots show a clear trend in favour of the stride-based models. But many of the plots show a substantial amount of variability.

These results justify our design choices for the slot attention based models to achieve the goal of discovering meaningful units which have the same level of abstraction as the previously proposed units. In addition, although our stride-based models fall behind the slot attention based models in many cases, they can be utilized as strong baselines for evaluating such unsupervised models and to find acceptable abstract units which can omit the need for tokenizing text as a preprocessing step.

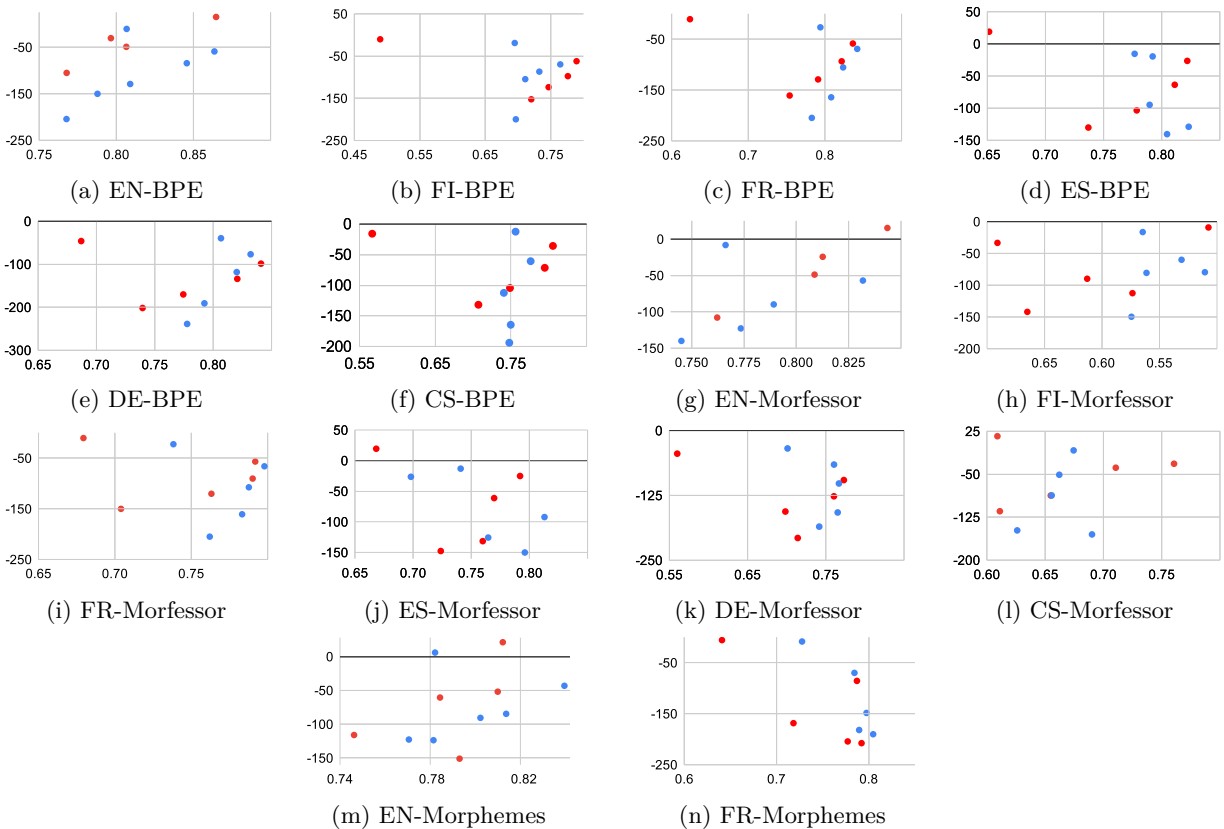

Figure 4: 2D graphs showing the informativeness trade-off for slot attention models (blue points) and the stride-based models (red points). The x-axis shows the F1 measure for the forward probe and the y-axis shows the negative log-density function loss for the reverse probe, so better models are lower and further to the right.

## 4.5 Downstream Task Evaluation

The focus of this work is developing the analogue of visual object discovery methods in the text domain, and developing an intrinsic evaluation framework for this line of work. We showed above that the information captured by our induced units are similar to the information in previously proposed representations, and thus there is every reason to believe that the demonstrable effectiveness of these predefined representations will also apply to our induced representations. To confirm the effectiveness of our induced meaningful units in downstream tasks, we report here results using our induced representations in a downstream task which we believe to be representative of their potential.

As our downstream task, we used the challenge set released by Hofmann et al. (2022), which they argue serves as a benchmark for pretrained language models' tokenizers. The task is to classify each ArXiv title into its corresponding sub-area (within a category of 20) in the three subjects of CS, Maths, Physics. The dataset requires a challenging generalization from a small number of short training examples with highly complex language (Hofmann et al., 2022). For this reason, we believe it is a good benchmark for evaluating our unit induction models.

In order to evaluate the quality of the induced units, we first freeze the representations we get from our trained models and then train classifiers for this downstream task. Since our models output a set of vectors as the high-level representation of the character sequence, we design an attention-based classifier to perform the task. In particular, after applying a shared 2 layer MLP (with ReLU non-linearity) on top of each of the representation's vectors, we apply attention over the resulting keys and values using a learnable query

|  | Dev | | | |
| --- | --- | --- | --- | --- |
|  | Cs | Maths | Physics | Avg |
| stride=1 | 0.3053 | 0.3315 | 0.3867 | 0.3411 |
| stride=6 | 0.3919 | 0.3976 | 0.468 | 0.4191 |
| BPE-based | 0.3196 | 0.3383 | 0.3693 | 0.3424 |
| slot-attn(r=6) | 0.3911 | 0.4229 | 0.4742 | **0.4294** |
|  | Test | | | |
|  | Cs | Maths | Physics | Avg |
| stride=1 | 0.2934 | 0.3181 | 0.3839 | 0.3318 |
| stride=6 | 0.3833 | 0.3863 | 0.4529 | 0.4075 |
| BPE-based | 0.3095 | 0.345 | 0.3578 | 0.3374 |
| slot-attn(r=6) | 0.397 | 0.424 | 0.4682 | **0.4297** |

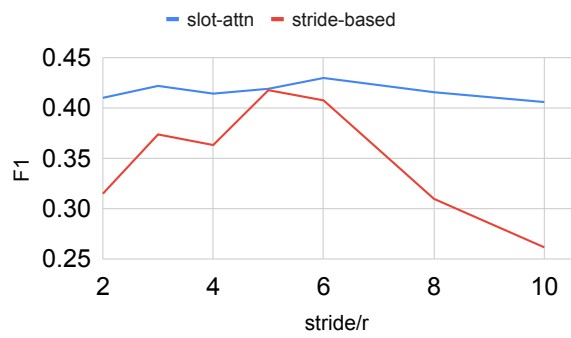

(a) Results with varying stride/$r$ on the test set.

Table 3: Arxiv-L classification results without fine-tuning the models.

vector. Then, we linearly project the resulting vector to compute the score for each sub-area, and apply softmax with cross-entropy loss. Following Hofmann et al. (2022), we use F1 as our evaluation metric and average across the subjects. We compare our Slot Attention model with Stride-based model with stride 6 and a character-based model (stride=1) in addition to a BPE-based model, as shown in Table 3.[5] In order to be fair in comparison, we train the BPE-based baseline model under the same setup as a character-based model (stride=1) by only modifying the inputs and targets to be BPE tokens of the sentence. These results indicate that the representations induced by our proposed models work much better than a character-based and BPE-based model on this challenging task, confirming that they could be a useful replacement for tokenization methods in downstream tasks. The slot attention representations also perform better than the stride-based representations. The BPE-based model does not work as well as other models and we believe this is due to the fact that BPE struggles to find good tokenization of the rare technical words found in ArXiv topics and usually splits them into meaningless tokens.

Since there is no target representation in this evaluation, we also consider varying the target percentage of vectors in the induced representations, shown in Figure 3a. Interestingly, the best performance is achieved with induced units which are around the same granularity as those used in the BPE target representation (stride=6). For the stride-based models, results are very dependent on finding the right hand-coded level of granularity, whereas our slot attention model is more robust to the choice of the target level of granularity. Perhaps this is due to the ability of its $L_0$Drop layer to adjust the number of units depending on the content of each individual example.

## 5   Conclusions

In this paper, we propose Dynamic Capacity Slot Attention for discovering meaningful units in text in an unsupervised fashion. We use an auto-encoder architecture for encoding a character sequence into a set of continuous slot vectors and decoding them to reconstruct the original sequence. We enforced the set of slots to act as a bottleneck in transferring information between the encoder and the decoder by adding a constant noise to their vectors and integrating an $L_0$ regularizing layer on top of the slots which only retains the necessary vectors. In addition, we propose a set of stride-based models which could serve as an alternative to our main model. We evaluate our model by probing the equivalence between the pruned slots and predefined tokens. In particular, we propose to do reverse probing as well as the normal way of probing. Our experiments show that our representations effectively capture meaningful information at a higher level of abstraction. Moreover, we show the usefulness of our induced units in a challenging classification task.

---

[5]Note that our numbers are not comparable to the ones reported in Hofmann et al. (2022) because we use a completely different setup (in terms of both pre-training data and architecture)

This work is a first step towards replacing hand-coded abstract units in text with representations induced within a deep learning architecture, demonstrating that it is feasible to improve over hand-coded units, even in the text domain. More generally, it addresses the issue of entity induction, which is a fundamental property of cognitive representations but is currently poorly understood. We believe that the proposed models and evaluation method will lead to greater progress on this fundamental problem.

**Future work.** One of the advantages of our unsupervised approach to inducing abstract units is that the method is not specific to characters, or indeed to any observable level of representation. Thus it can be trained in an end-to-end fashion with different training objectives, it can be used to induce any level of abstraction, and it can be stacked to induce multiple levels of abstraction. This could potentially avoid the need to hand-code different levels of abstraction, such as vectors associated with tokenization and sentence markers. Replacing hand-coded model design choices with choices which are induced within a deep learning architecture is a central objective of this work.

**Limitations.** We learn a set of abstract continuous vector representations of the input and we do not explicitly discover morphemes or morpheme segments. Although the visualized attention maps show interesting patterns of input segmentation (see Figure 2), the vertical bands are fading near the end-points. More specifically, it is not straight-forward to determine boundaries between the bands as the transition is done smoothly due the continuous nature of the attention function. For this reason, we could not obtain good segmentations by employing simple heuristics on the attention maps.

**Reproducibility Statement.** We have completely explained the details of our experiments in Appendix B. It includes the data we have used and how we have processed it, in addition to the models' parameters and training details.

## Acknowledgments

We would like to thank Florian Mai, Andreas Marfurt, Andrei Catalin Coman, and Fabio Fehr for their valuable input and discussions at various stages of this project. We would like to extend our thanks to the anonymous reviewers for their constructive feedback and insights, which significantly contributed to the improvement of this work. Melika Behjati was funded by the Swiss National Centre of Competence in Research (NCCR) under the project Evolving Language, grant number "51NF40_180888".

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

# A Supplementary Results

## A.1 Slot Initialization Analysis

Table 4: Forward probing results of different slot initializations for Spanish language.

| | | BPE | | | Morfessor | | |
|---|---|---|---|---|---|---|---|
| Init. | Model | P | R | F1 | P | R | F1 |
| eq. (2) | untrained | 0.71 | 0.09 | 0.17 | 0.71 | 0.11 | 0.19 |
| (ours) | slot-attn | 0.96 | 0.73 | **0.82** | 0.95 | 0.74 | **0.83** |
| | | | | | | | |
| eq. (1) | untrained | 0.80 | 0.10 | 0.18 | 0.78 | 0.12 | 0.20 |
| | slot-attn | 0.97 | 0.51 | 0.66 | 0.96 | 0.51 | 0.66 |

We compare our proposed initialization with the original definition of Slot Attention, where slots are randomly initialized from a single shared distribution. Table 4 shows the probing results of our model under different slot initializations. Having a separate $\mu$ for each slot (Equation (2)) increases the capacity of the model and thus yields better results in comparison to sharing $\mu$ between the slots. The model especially achieves higher recall in this case, which implies better recovery of the BPE and Morfessor tokens as it can accurately model a greater number of units.

## A.2 Visualization of the Attention Maps

### A.2.1 Attention of Slots over the Input

In Figures 5 and 6, we illustrate the attention of slots over the input as well as the attention of decoder over the slots for the same input. The two attention maps show similar patterns, but on the decoder side the attention weights are higher and therefore the patters are more visible. We believe that at each generation step, the decoder only attends to the slots which contain the information about that specific character and thus, the attention patterns are stronger at decoding time. For these reason, we used the attention of the decoder over the slots in section 4.3. Interestingly, in Figure 6b, there are some slots which are only attending to the space boundaries (the horizontal bands).

### A.2.2 Decoder Attention for the Stride-Based models

Figure 7 visualizes the attention of decoder over the vectors of different stride-based models. As expected, the vertical bands are often of the same length and too much overlapping for larger strides. As a result, the bands do not correspond to meaningful units in the input in contrast to the slot attention based model (see Figure 2c).

## A.3 Forward Probing Examples

Table 5 shows two examples of the probing classifiers' predictions given the learned slots. As explained in 4.4, the model is quite precise in predicting *non-empty* labels.

## A.4 Bidirectional Probing Results' Tables

Tables 6 to 16 show the detailed results of the performance of forward and reverse probing tasks visualized in Figure 4 (Reverse reconstruction vs F1). n_input denotes the input length which is 128 in our experiments.

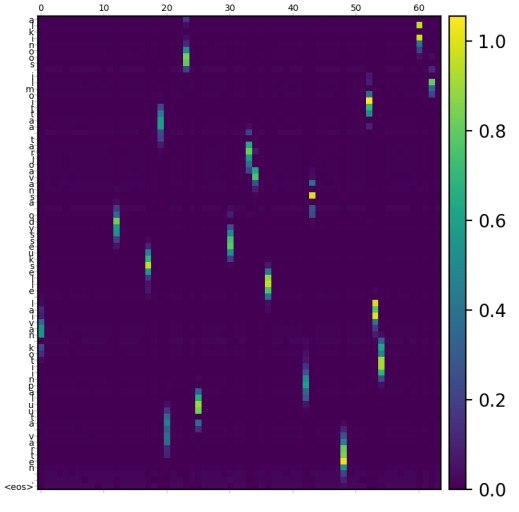

(a) Attention of decoder (y-axis) over slots (x-axis).

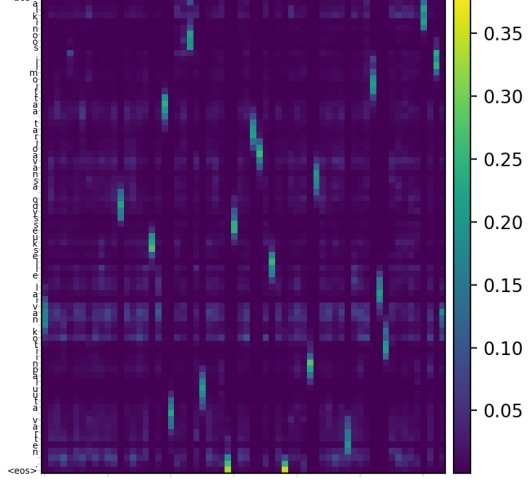

(b) Attention of slots (x-axis) over the input (y-axis) before the $L_0$Drop.

Figure 5: Illustration of the Attention of Slots over the input vs the Attention of decoder over the slots for Finnish language.

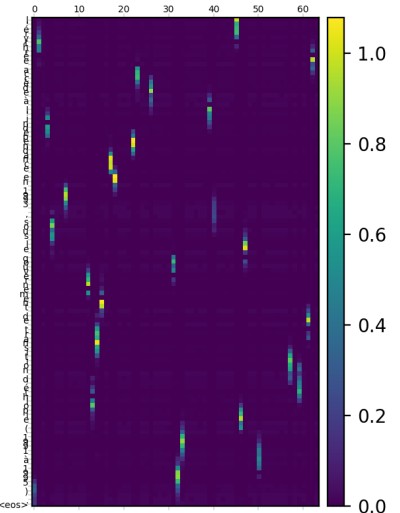

(a) Attention of decoder (y-axis) over slots (x-axis).

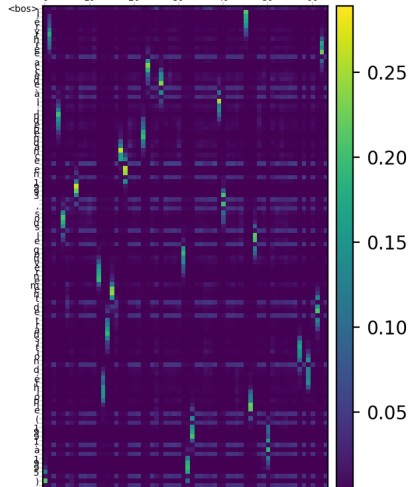

(b) Attention of slots (x-axis) over the input (y-axis) before the $L_0$Drop.

Figure 6: Illustration of the Attention of Slots over the input vs the Attention of decoder over the slots for French language.

# B  Settings

## B.1  Data

As for the datasets, we lowercased the text and retained the characters which occur more than 25 times in the corpus, following Kawakami et al. (2017). We replace the low-frequent characters with an unknown placeholder. Table 18 shows the licenses of each dataset that we used in our experiments. We used the same train/validation/test splits as provided in the mentioned datasets.

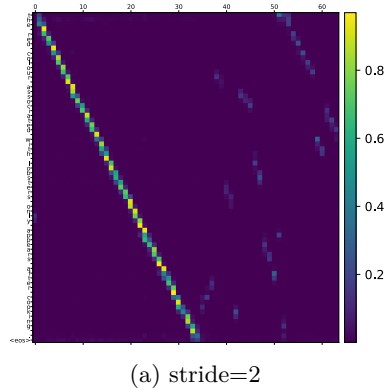
(a) stride=2

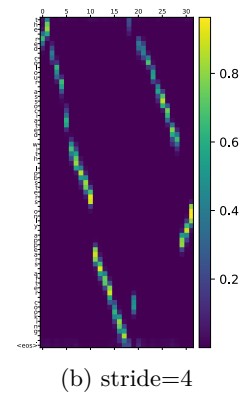
(b) stride=4

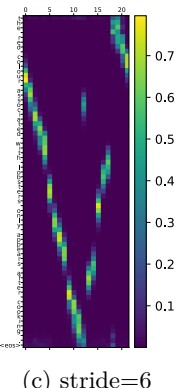
(c) stride=6

Figure 7: Attention of decoder over the stride-based model vectors (x-axis) while generating every character (y-axis). The target sentence is "*the red colour associated with lobsters only appears after cooking.*".

Table 5: Input and output pairs from Spanish and German datasets. The predictions are sorted based on their matching target. The empty label is shown as '#' and wrong predictions are shown in red.

| input | probing classifier prediction |
|---|---|
| una razón de su auge fue su aparente éxito en tratar enfermos por epidemias infecciosas . | [BPE] una razón de su auge fue su aparente éxito en tratar enfermos por epi#as# inf#ocosasón
[Morfessor] una razón de su auge fue su aparente éxito en tratar enfermcio por epidemias#s .rs |
| außerdem wurde er zum besten spieler des turniers gewählt . | [BPE] außerdem wurde er zum sten spieler des #ur#ierers gewähl# .
[Morfessor] außerdem wurde er zutur besten spieler des turniers gewählt # |

Table 6: Stride-based models for English with BPE targets

| stride | stride=2 | stride=3 | stride=4 | stride=6 |
|---|---|---|---|---|
| F1 | 0.8647 | 0.7965 | 0.8065 | 0.7678 |
| Loss(1e+5) | 1.318 | 1.7084 | 2.2163 | 3.4569 |
| P | 0.9376 | 0.9125 | 0.9236 | 0.9514 |
| R | 0.8106 | 0.7335 | 0.732 | 0.6667 |
| Acc | 0.9114 | 0.8923 | 0.8811 | 0.8596 |
| Reverse reconstruction | 15.14 | -30.44 | -49.04 | -105.1 |

Table 7: Slot Attention based models for English with BPE targets

| limit $(r)$ | r=2 | r=3 | r=4 | r=5 | r=6 | r=8 |
|---|---|---|---|---|---|---|
| F1 | 0.8067 | 0.8636 | 0.8457 | 0.809 | 0.7878 | 0.7677 |
| Loss(1e+5) | 1.5364 | 1.2951 | 1.5757 | 2.0522 | 2.3737 | 2.8781 |
| P | 0.8634 | 0.924 | 0.9394 | 0.9555 | 0.9635 | 0.9763 |
| R | 0.7687 | 0.8213 | 0.7819 | 0.7196 | 0.6863 | 0.6517 |
| Acc | 0.8845 | 0.9139 | 0.9058 | 0.8934 | 0.8865 | 0.8798 |
| Reverse reconstruction | -10.56 | -58.98 | -84.32 | -129 | -150.1 | -204.4 |

Table 8: Stride-based models for English with Morpheme targets

| stride | stride=2 | stride=3 | stride=4 | stride=5 | stride=6 |
|---|---|---|---|---|---|
| F1 | 0.8108 | 0.8086 | 0.7835 | 0.792 | 0.7461 |
| Loss(1e+5) | 2.6374 | 2.5115 | 2.6292 | 2.7607 | 3.5556 |
| P | 0.8971 | 0.883 | 0.8812 | 0.9083 | 0.9155 |
| R | 0.7479 | 0.7567 | 0.7179 | 0.7179 | 0.6498 |
| Acc | 0.8973 | 0.8964 | 0.8833 | 0.8897 | 0.8688 |
| Reverse reconstruction | 21.26 | -51.9 | -60.54 | -151 | -116 |

Table 9: Slot attention based models for English with Morpheme targets

| limit $(r)$ | r=2 | r=3 | r=4 | r=5 | r=6 | 8 |
|---|---|---|---|---|---|---|
| F1 | 0.7821 | 0.8394 | 0.8136 | 0.8022 | 0.7814 | 0.7705 |
| Loss(1e+5) | 2.0958 | 1.7118 | 2.1506 | 1.9883 | 2.3083 | 2.4825 |
| P | 0.8353 | 0.8844 | 0.8952 | 0.9083 | 0.9157 | 0.9331 |
| R | 0.7482 | 0.8101 | 0.7577 | 0.7383 | 0.7039 | 0.6789 |
| Acc | 0.884 | 0.9116 | 0.9017 | 0.9025 | 0.896 | 0.8949 |
| Reverse reconstruction | 5.975 | -43.17 | -84.65 | -90.6 | -123.7 | -122.7 |

Table 10: Stride-based models for English with Morfessor targets

| stride | stride=2 | stride=3 | stride=4 | stride=6 |
|---|---|---|---|---|
| F1 | 0.8435 | 0.8126 | 0.8087 | 0.7622 |
| Loss(1e+5) | 2.7637 | 4.3381 | 2.6835 | 3.7541 |
| P | 0.9175 | 0.9012 | 0.9162 | 0.9458 |
| R | 0.7905 | 0.7529 | 0.7388 | 0.6606 |
| Acc | 0.9026 | 0.8872 | 0.8842 | 0.8595 |
| Reverse reconstruction | 15.28 | -24.36 | -48.84 | -107.9 |

Table 11: Slot attention-based models for English with Morfessor targets

| limit $(r)$ | r=2 | r=3 | r=4 | r=5 | r=6 |
|---|---|---|---|---|---|
| F1 | 0.7662 | 0.8318 | 0.7892 | 0.7735 | 0.7451 |
| Loss(1e+5) | 2.7095 | 1.9951 | 2.7874 | 2.728 | 3.1524 |
| P | 0.8533 | 0.9089 | 0.9128 | 0.9323 | 0.9367 |
| R | 0.7091 | 0.7787 | 0.7087 | 0.6793 | 0.6383 |
| Acc | 0.8636 | 0.8975 | 0.8777 | 0.8763 | 0.8668 |
| Reverse reconstruction | -8.193 | -57.02 | -89.83 | -122.8 | -140 |

Table 12: Stride-based models for French with BPE targets

| stride | stride=2 | stride=3 | stride=4 | stride=5 | stride=6 |
|---|---|---|---|---|---|
| F1 | 0.6239 | 0.8365 | 0.8219 | 0.7911 | 0.754 |
| Loss(1e+5) | 5.3584 | 2.1264 | 2.669 | 3.6093 | 4.5231 |
| P | 0.8814 | 0.921 | 0.9356 | 0.9485 | 0.9584 |
| R | 0.4916 | 0.7756 | 0.7454 | 0.6939 | 0.6399 |
| Acc | 0.7767 | 0.885 | 0.8769 | 0.8603 | 0.8392 |
| Reverse reconstruction | -10.51 | -58.4 | -93.31 | -129 | -160.7 |

Table 13: Slot attention-based models for French with BPE targets

| limit $(r)$ | r=2 | r=3 | r=4 | r=5 | r=6 |
|---|---|---|---|---|---|
| F1 | 0.7941 | 0.8422 | 0.8238 | 0.8082 | 0.7829 |
| Loss(1e+5) | 2.0313 | 1.8495 | 2.3189 | 3.0405 | 3.7684 |
| P | 0.8694 | 0.9247 | 0.934 | 0.964 | 0.9786 |
| R | 0.7403 | 0.7826 | 0.7488 | 0.7087 | 0.665 |
| Acc | 0.8594 | 0.8876 | 0.8789 | 0.8725 | 0.8593 |
| Reverse reconstruction | -26.5 | -68.98 | -105.4 | -164.2 | -204.7 |

Table 14: Stride-based models for French with Morpheme targets

| stride | stride=2 | stride=3 | stride=4 | stride=5 | stride=6 |
|---|---|---|---|---|---|
| F1 | 0.6409 | 0.7872 | 0.7922 | 0.7772 | 0.7184 |
| Loss(1e+5) | 5.5099 | 3.1225 | 3.0613 | 3.6183 | 4.8101 |
| P | 0.9014 | 0.8906 | 0.8936 | 0.914 | 0.9281 |
| R | 0.5051 | 0.714 | 0.722 | 0.6885 | 0.5989 |
| Acc | 0.8107 | 0.872 | 0.8746 | 0.8682 | 0.8438 |
| Reverse reconstruction | -5.61 | -85.57 | -207.7 | -204.3 | -168.5 |

Table 15: Slot attention-based models for French with Morpheme targets

| limit $(r)$ | r=2 | r=3 | r=4 | r=5 | r=6 |
|---|---|---|---|---|---|
| F1 | 0.7277 | 0.7845 | 0.7976 | 0.8048 | 0.7896 |
| Loss(1e+5) | 3.5534 | 2.9845 | 2.835 | 2.565 | 3.0282 |
| P | 0.8403 | 0.876 | 0.8889 | 0.9195 | 94.05 |
| R | 65.26 | 0.7195 | 0.7329 | 0.7269 | 0.6921 |
| Acc | 0.8406 | 0.8679 | 0.8762 | 0.8829 | 0.8777 |
| Reverse reconstruction | -8.35 | -69.87 | -148.4 | -190.2 | -181.9 |

Table 16: Stride-based models for French with Morfessor targets

| stride | stride=2 | stride=3 | stride=4 | stride=5 | stride=6 |
|---|---|---|---|---|---|
| F1 | 0.6794 | 0.7919 | 0.7902 | 0.7632 | 0.704 |
| Loss(1e+5) | 5.5356 | 3.6372 | 3.7099 | 4.5012 | 5.8283 |
| P | 0.9041 | 0.9086 | 0.9189 | 0.9368 | 0.9454 |
| R | 0.5533 | 0.7117 | 0.7045 | 0.6578 | 0.5738 |
| Acc | 0.8078 | 0.8615 | 0.8619 | 0.8481 | 0.821 |
| Reverse reconstruction | -10.33 | -56.74 | -90.33 | -120.3 | -150.1 |

Table 17: Slot attention-based models for French with Morfessor targets

| limit $(r)$ | r=2 | r=3 | r=4 | r=5 | r=6 |
|---|---|---|---|---|---|
| F1 | 0.7383 | 0.7979 | 0.7877 | 0.7833 | 0.7621 |
| Loss(1e+5) | 4.2044 | 3.1744 | 3.4661 | 3.4707 | 4.229 |
| P | 0.867 | 0.9016 | 0.9117 | 0.9466 | 0.963 |
| R | 0.6474 | 0.7256 | 0.7044 | 0.681 | 0.6425 |
| Acc | 0.8302 | 0.8646 | 0.8602 | 0.8619 | 0.8521 |
| Reverse reconstruction | -22.51 | -66.06 | -107.5 | -160.8 | -205 |

Table 18: Data licenses

| dataset | license |
|---------|---------|
| WikiText2 (Merity et al., 2017) | Creative Commons Attribution-ShareAlike 3.0 Unported License (link to dataset) `https://aclanthology.org/P17-1137/` |
| Multilingual Wikipedia Corpus (MWC) (Kawakami et al., 2017) MorphoLex (Sánchez-Gutiérrez et al., 2018; Mailhot et al., 2020) | Creative Commons Attribution-NonCommercial-ShareAlike 4.0 License (CC BY-NC-SA 4.0) (`https://lindat.mff.cuni.cz/repository/xmlui/handle/11234/1-4629#`) |

Table 19: Hyperparameters of the main model.

| module | parameter | value |
|--------|-----------|-------|
|  | batch size | 16 |
|  | learning rate | $1e-4$ |
| transformer | model dimension | 256 |
| transformer | feedforward layer dimension | $4 \times 256$ |
| transformer | dropout rate | 0.1 |
| $L_0$Drop | $\beta$ | 0.66 |
| $L_0$Drop | $\epsilon$ | 0.1 |
| Slot Attention | Slot dimension ($D_{slots}$) | 128 |
| Slot Attention | MLP hidden dimension | $2 \times D_{slots}$ |
| Slot Attention | GRU hidden dimension | $D_{slots}$ |
| Slot Attention | $\delta$ | $1e-8$ |
| Slot Attention | $T$ | 1 |

## B.2 Main Model Settings

Table 19 shows the remaining list of hyperparameters in training the main model. We scheduled the $\lambda$ parameter in the training loss to start with a low value of $2 \times 10^{-5}$ and exponentially increase it every 10 epochs until it reaches a certain limit. In particular, we schedule the $\lambda$ to exponentially increase with ratio 2 for English until reaching $6.4e-4$ and 1.5 for the rest of languages. More specifically, we stop the exponential increase after reaching $3e-4$ for German and Spanish and $5e-4$ for French, Finnish and Czech. We tried the stopping thresholds ranging from $3 \times 10^{-4}$ to $6 \times 10^{-4}$. These scheduling values lead to having roughly as many slots as the average number BPE units per sentence. We also tried our model with statistic $\lambda$ in the $\{10^{-4}, [1, 3, 5, 6, 7, 8] \times 10^{-5}\}$ which did not lead to stable results at training time. We tried 16, 32 and 64 slots in our experiments. We chose the number of slots to be half of the maximum sequence length (128) as this is a reasonable upperbound which also matches the maximum number of BPE or Morfessor units. We tried the transformer encoder with 4 and 6 layers but qualitatively did not find any improvements. We run the Slot Attention algorithm for $T$=1 iterations. We choose $T$=1 iterations for simplicity and efficiency, and because preliminary experiments showed no improvements with more iterations. We leave the investigation of how to get improvements from more iterations to future work. Finally, we used Adam optimizer (Kingma & Welling, 2013) for training our models with learning rate $10^{-4}$ and train our models for 200 epochs.

## B.3 Forward Probe Settings

We train a probing classifier for mapping a slot's vector to the target representation's vocabulary, namely, $f(m'_i) : \mathrm{R}^{D_{slots}} \to \mathrm{R}^S$, where $S$ is the number of tokens of the target representation. We apply the classifier with shared parameters over each of our slots and obtain a *set* of predictions, i.e., $\{f(m'_1), f(m'_2), \ldots, f(m'_K)\}$. As we are dealing with a set, during training we need to find a one-to-one matching between the classifier's predictions and the target tokens. Therefore, we use the Hungarian matching algorithm Kuhn (1955) for finding the best match in terms of minimizing the classification loss. Consider the best matching as $i_j \to j$,

which matches the $i_j$th slot with the $j$th output (i.e., $y_j$). We then compute the loss as $\mathcal{L} = \sum_{j=1}^{K} l(f(m'_{i_j}), y_j)$, where $l$ is the cross-entropy between the predictions and targets.

Our probing classifier consists of two fully connected layers with ReLU activation function in between the two layers. The hidden dimension of the classifier is the same as the slots' dimension, which is 128.

We use the same datasets as our main model for training and testing the probes. We train BPE with a vocabulary size of 5000 for all languages. For Morfessor, we use the pretrained model and consider the set of its outputs on the training data as our target representation. As for the morpheme targets, we take the morphemes for the words which were available in the linguistically annotated data (i.e., MorphoLex (Sánchez-Gutiérrez et al., 2018; Mailhot et al., 2020)) and for the rest of the words we take Morfessor outputs as an approximation of morphemes.

For training the probing classifiers we take a batch size of 4, since the Hungarian matching algorithm requires a huge amount of memory. We train our classifiers for 200 epochs with Adam optimizer with learning rate of $1 \times 10^{-3}$.

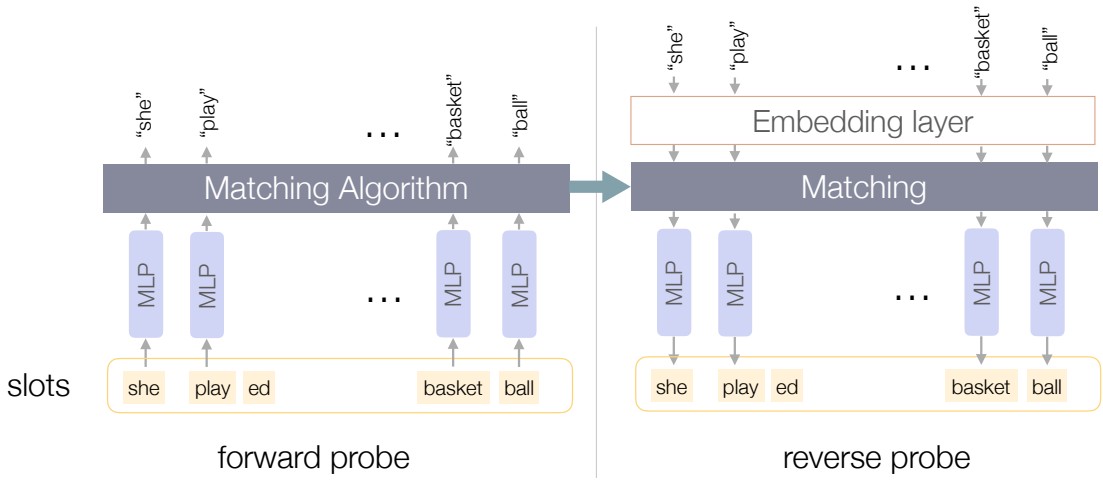

Figure 8: An illustration of forward and reverse probing.

## B.4 Reverse Probe Settings

We illustrate how forward and reverse probing are related to each other in Figure 8. We first assign a one-hot vector to the tokens in the target set of units (i.e., $R^S$, where $S$ is the target set size). Then, we learn an embedding layer to map every one-hot vector into a continuous vector with dimension 128. Afterward, we pass the embedded token into two fully connected layers with ReLU activations in between with the hidden dimension 128. We then predict the mean and the standard deviation of a Gaussian distribution. The dimension of the Gaussian distribution $d$ is $D_{slots} = 128$. We use the same matching between the target units and slots as in the forward probing. Namely, we do not run the matching algorithm for this experiment. As for the training objective, we minimize the negative log-likelihood of the slot vector given this distribution, i.e., $-\log(p(m_i|\mu_{\text{predicted}}, \sigma_{\text{predicted}}))$ which is equivalent to minimizing $\sum \frac{1}{2\sigma^2_{\text{predicted}}}(m_i - \mu_{\text{predicted}})^2 - \log(\sigma_{\text{predicted}})$. We train this model with Adam optimizer and the learning rate $1 \times 10^{-4}$ for 200 epochs. We report the best evaluation loss on test set.

## B.5 Arxiv Classifier Settings

The hidden dimension of the MLP and also the query, key and value matrices are set to the same dimension as the slots, namely, 128. We train the classifier with a batch size of 32 for 200 epochs with Adam optimizer

Table 20: List of packages and their versions.

| package | version | use |
|---|---|---|
| NLTK | 3.5 | sentence and word tokenization |
| youtokentome | 1.0.5 | BPE implementaion |
| polyglot | 16.7.4 | morfessor implementation |
| matplotlib | 3.3.2 | attention maps visualization |
| scipy | 1.2.2 | hungarian algorithm implementation |
| numpy | 1.19.1 | |

with a learning rate of $1e-4$. For the Arxiv-L dataset each subarea has 1000 samples which would be 20000 samples in total.

## C   Infrastructure

We use PyTorch version 1.2.0 framework and Python version 3.6.9 for implementing our code. Table 20 shows the rest of the libraries we use. We run our code on a single GPU with model GTX1080ti and the operating system Debian10 (Buster) 64-bit. We use the same compute for all of our experiments including training the models and probes. The training time for the main model is five hours and for the probings is around 2 days. For reporting each of the results we run our algorithm once, since it would be too computationally expensive.

