# OpenReview forum: "Inducing Meaningful Units from Character Sequences with Dynamic Capacity Slot Attention"
_TMLR — Accepted by TMLR_

### Review · Reviewer_21YV · 2023-05-30

**Summary Of Contributions:**

This paper proposes Dynamic Capacity Slot Attention for discovering meaningful units in text in an unsupervised fashion. The method is based on an auto-encoder architecture for encoding a character sequence into a set of continuous slot vectors and decoding them to reconstruct the original sequence. The authors enforced the set of slots to act as a bottleneck in transferring information between the encoder and the decoder by adding a constant noise to their vectors and integrating an $L_{0}$ regularizing layer on top of the slots which only retains the necessary vectors. The proposed method is evaluated by both the probing task and a downstream classification task.

**Audience:**

Yes

**Broader Impact Concerns:**

Although I find this work interesting, it does not necessarily demonstrate the advantages over existing studies.

**Claims And Evidence:**

Yes

**Requested Changes:**

More thorough comparisons are needed, e.g., comparison with existing approaches, and more comprehensive experiments on downstream tasks.

**Strengths And Weaknesses:**

Strengths:

1. This work is novel at replacing explicit language unit segmentation with representations induced within a deep learning model. In previous studies, language unit segmentation is often conducted using a statistical approach to count the word concurrence with a (small) fixed context window. In contrast, this work does not explicitly segment a sentence into units and considers the whole sentence as the context.

2. The evaluation is based on both the probing task and downstream classification task, which verifies the effectiveness of the proposed method. The induced units tend to capture meaningful information at an appropriate level of abstraction.

Weaknesses:

1. Although the authors have discussed the connections and differences with previous morphology learning and tokenization approaches in Section 3, the advantage of the proposed method is not clear. As previous methods will segment a sentence into explicit tokens,  those explicit tokens can serve as the fundamental part of any deep learning model. A more fair baseline is a same-sized Transformer encoder fed with tokens segmented by public tokenization approaches (e.g., SentencePieces [1]):

**Baseline**: sentence -> [SentencePiece] -> subwords -> [Transformer Encoder] -> [Classification]

**The Proposed Model**:  sentence -> [The (Trained) Proposed Encoder] -> [Classification]

The evaluation can be based on any downstream tasks (e.g., the classification task in the paper), to show that the proposed approach is indeed a more effective alternative to the existing paradigm of "tokenization + encoding."

2. The task formulation may be misleading.

"The unsupervised induction of abstract meaningful units from sequences of characters is a novel task which requires both novel methods and novel evaluations."

- According to the problem formulation in Section 2.1, the task seems to be the same as the standard language units segmentation tasks in existing studies [2]. Besides, as the proposed approach does not explicitly produce the set of morphemes-like units, the example output {she, play, -ed, basket, -ball} might not be accurate for indicating the real output of the model. Maybe it is better to highlight the major difference in "abstract".

Minor Comments:

The position of Section 3 Related Work is weird. Why not place it before introducing the proposed approach?

[1] Kudo, T. and Richardson, J., 2018, November. SentencePiece: A simple and language independent subword tokenizer and detokenizer for Neural Text Processing. In Proceedings of the 2018 Conference on Empirical Methods in Natural Language Processing: System Demonstrations (pp. 66-71).

[2] Zhang, Z., Zhao, H., Ling, K., Li, J., Li, Z., He, S. and Fu, G., 2019. Effective subword segmentation for text comprehension. IEEE/ACM Transactions on Audio, Speech, and Language Processing, 27(11), pp.1664-1674.

---

> ### Author Response · Authors · 2023-08-15
> **Response to Reviewer 21YV**
>
> Thank you for your insightful review of our work.
>
> 1- Thank you for suggesting the experiment. Please refer to the general response for a complete discussion of the proposed baseline.
>
> 2- We clarified the task formulation (Section 2.1) to make it explicit that we want to learn a set-of-vector representation of a text.  This makes the task clearly different from morphology induction or string segmentation, and better suited as part of a deep learning model.  We still expect each vector in the set to correspond to something like one morpheme in the text, since the set of vectors should represent the set of meaning-bearing units in the text.  But the vectors are computed dynamically, without assuming a fixed vocabulary of morphemes.
>
> Minor comment: We thought placing the related work as the second section would interrupt the reader from following the paper’s story and it would allow them to have a better view of the related work once they understood the method.

---

### Review · Reviewer_dFr6 · 2023-06-15

**Summary Of Contributions:**

This paper presents a way to induce meaningful units from character sequence by adapting slot attention model. They evaluate their model through a suite of experiment, such as (1) reconstructing the original text, (2) probing with other character unit discovery methods and (3) downstream application.

**Audience:**

Yes

**Claims And Evidence:**

Yes

**Requested Changes:**

* Strengthen discussion of related work (see weakness).
* The current downstream evaluation is pretty weak in that it only compares to its own baseline. It should compare to other methods such as BPE.
* I'm a bit unclear about the efficiency of this model, especially compared to its stride-based baseline. Adding a discussion on this would be helpful.
* Visualization/analysis of learned slot patterns would be helpful. Is it more likely to assign consecutive characters to the same slots? The current visualization (Figure 2) is on decoder attention to slots, but you can also compute / analyze the attention over slots for each character?


**Strengths And Weaknesses:**

Strength: * The paper is clearly written and well organized. It addresses interesting task of inducing meaningful text units in an unsupervised fashion.
* The approach of adapting slot attention (originally used in vision) for text is interesting and shows promising empirical results.
* The experiments span multiple languages, and claims are mostly reasonable.

Weakness: * Relate work is weak. They only have one line discussing char former (Tay et al) and canine (Clark et al), but the motivation is fairly similar, and they should be discussed more throughly. Also, HuBERT model (from speech community) is also quite relevant in discovering units from self-supervision.

* Comparison: They report probing results with BPE/Morfessor and Morpheme. But I think comparing to word-piece will be helpful, as they are two different, popular mechanisms, especially given the result of [Bostrom and Durrett EMNLP20] argues workpiece model recovers more morphologically aligned subword units. Wordpiece is also slightly more relevant to proposed approach, as it is derived from learned language model similar to autoencoder here.

* Current downstream task evaluation is pretty weak. See requested change.

* It would be really interesting to train a small scale LM with units discovered from the proposed method and compare their performances (perplexities). As the discovered units are continuous and do not map to character sequence, this is not straightforward, but discussing this will be nice.

---

> ### Author Response · Authors · 2023-08-15
> **Response to Reviewer dFr6**
>
> We thank you for your time and effort in reviewing our work and appreciate your feedback.
>
> >Related work is weak.
>
> We updated the related work section to include a discussion on Charformer and CANINE and also the unit discovery work in speech in the new version of the paper.
>
> >Comparison with Wordpiece.
>
> Thank you for your suggestion. While wordpieces are better aligned to morphological features than BPEs, we think including them would not add value to our probing evaluations as we are already comparing our units to real gold morphemes. Moreover, the wordpiece algorithm is using a unigram language model where no neural network is involved. However, we can perform this evaluation if the reviewer finds it crucial.
>
> >Downstream evaluation is pretty weak.
>
> We have performed the downstream task with a BPE-based model and added the details to the new version of the paper (Section 4.5). Please refer to the general response for a discussion.
>
> >Discussion on small-scale LM.
>
> Thank you for your suggestion. This would be an interesting future direction for our work.
> The trained decoder can go from slots to characters, but we don't have a prior over slots so we can't compute LM perplexities.  Maybe sampling could be done by training some form of diffusion model over slots, but that still doesn't give us perplexities.  This would be a really interesting problem, but it is way outside the scope of this work.
>
> >Discussion on the efficiency of the models.
>
> We will add a quantitative discussion on the efficiency of the models in the next version of the paper. Intuitively, we think that slot attention will be slightly slower than stride-based models, with the benefit of discovering better units.  Compared to character-based models, there should be a speedup in cross-attention due to fewer units.
>
> >Visualization/analysis of the learned slot patterns.
>
> Evaluation of unsupervised learning is always difficult, but we have analyzed the relation of the induced representations to previous representations through probing, and we have visualized the model’s attention patterns.  We have also added an analysis (Section 4.3.1) of the average attention pattern for each slot, showing that they correlate with position.  Regarding visualization,  Figure 2 provides attention over slots for each character being generated by the decoder, and Figures 5 and 6 (in the Appendix A.2) compare this to the attention over characters in the input for each slot.  The attention of the decoder over the slots shows which slots are being attended by the decoder while generating each character in the output. This shows which slots are carrying the information about that specific character.  The attention of slots over the input characters is less clear because contextualized character tokens convey more than just individual characters.  We would appreciate it if the reviewer could clarify what is exactly missing in the current visualizations and analysis.

---

### Review · Reviewer_JWZR · 2023-08-02

**Summary Of Contributions:**

The paper proposes an extension of Slot Attention to the text domain. To cope with the large vocabulary and variable sequence length, the paper proposes a new slot initialization and a method for dynamically choosing the slots. The paper also proposes a bi-directional probing method to evaluate whether the learned representations contain equivalent information as existing tokenization and morphology learning methods (e.g., BPE, Morfessor). The experiments are done across multiple languages, with a stride-based model as the baseline. Results show that the proposed model achieves better probing scores and higher downstream classification accuracy than the baseline.

**Audience:**

Yes

**Claims And Evidence:**

Yes

**Requested Changes:**

I think it is critical to provide some empirical evidence directly showing the advantage or similar performance of the proposed method over tokenization / morphology learning methods.

---

I thank the authors for adding the BPE baseline, clarifications, and related work. My concerns have been addressed.

**Strengths And Weaknesses:**

Strengths

The idea to extend Slot Attention from image to text is interesting. The paper proposes several adaptations of the original Slot Attention to cope with text inputs. The method makes sense and is easy to follow. Visualization shows that the method can achieve unsupervised grouping of characters.

Weaknesses
- My major concern is that there is insufficient evidence for the claim that the proposed method "could be a useful replacement for tokenization methods in downstream tasks".
    - There is no experiment comparing the proposed method with tokenization methods such as BPE in downstream tasks.
    - The only comparison is done with the stride-based baseline, and it seems unclear whether this is a strong baseline, and whether outperforming this baseline is a meaningful result.
    - The probing evaluation may provide some evidence, but it is unclear whether the probing score correlates well with downstream performance.
- For completeness, it would be better to provide more explanation/equation/figure of the forward/reverse probing method. It is a bit confusing why the forward probing is a classification problem, how empty labels are obtained, and what loss is used exactly.
- Related work could include more recent work on object-centric learning, such as [SLATE](https://arxiv.org/abs/2110.11405) which also uses a Transformer decoder.

---

> ### Author Response · Authors · 2023-08-15
> **Response to JWZR**
>
> Thank you for taking the time to review our paper and providing valuable feedback.
>
> >Insufficient evidence for the claim that the proposed method "could be a useful replacement for tokenization methods in downstream tasks".
>
> The wording “could be” and “potential” which we always use in the context of this issue clearly indicates that we are not making a claim to have demonstrated usefulness in general.  We have claimed to show that our method is a useful replacement for tokenization for one indicative downstream task.  However, we agree with the reviewer that this argument has been made much clearer and stronger by including a direct comparison to a BPE-based baseline.  Please refer to the general response for a discussion of this addition.  To answer the reviewer’s questions: the BPE-based baseline is much worse than the proposed model, and the stride-based baseline is a very strong baseline for this task.
>
> >Provide more explanation/equation/figure of the forward/reverse probing method.
>
> We have updated Appendix B.3 and B.4 to include a more precise explanation of the forward/backward probing setup, and Figure 8 provides an illustration of how the two probing methods are related. To answer your questions, in the forward probing setup, we treat every target unit as a label. More precisely, we have a fixed-size vocabulary for the target units where we add the empty label as an extra target. The empty label could be considered as a ‘no object’ label in the object detection scenario. We then train the classifier with cross-entropy loss.
>
> >Related work could include more recent work.
>
> Thank you for your suggestion. We have updated the related work section to include more recent work on the object-discovery literature.

---

### Review · Reviewer_e62S · 2023-08-03

**Summary Of Contributions:**

This submission studies the problem of finding "meaningful units" in text as represented by a sequence of characters in an unsupervised manner. These units are compared to "objects" in a visual scene (and their method is built upon prior work in the computer vision domain called slot attention). The proposed approach consists of learning to reconstruct the input sequence of characters by first encoding it into slots (queries in the transformer encoder attention) and then decoding the slots into the original input. The learned slots are evaluated in a few different ways, including a proposed method called reverse probing, over a few European languages. The results are compared primarily to a baseline fixed stride method, two subword tokenization methods from prior works, BPE and Morfessor, and annotated morphemes. The proposed method seems novel.

**Audience:**

Yes

**Broader Impact Concerns:**

Adequate.

**Claims And Evidence:**

No

**Requested Changes:**

1. I think the studied problem is very interesting but also conceptually challenging. It would help to devote more of the paper to explaining the *mathematical* model of meaning that you are pursuing and avoid over-usage of vague terms like "a higher level of abstraction." For example, does the ordering of the slots matter (for decoding)?
1. Relatedly, besides enhancing formalism of the presentation, it would also help to examine a few learned slots closely in the main text to provide support to some of claims you have about what the slots *can* represent ("such as ones that occur at different positions, or ones that have different types of forms" from section 2.2). For example, does the "-ing" slot you mentioned persistently activate for different input sentences? How about when "ing" appears in text not as a tense modifier, like in "mingle"?
1. A suggestion. You mentioned that vanilla slot attention was studied on synthesized data. Would evaluating on an artificial language help clarify what learned slots represent? What about programming languages?
1. A typo in section 2.2: "... by slots acting as queries..." -> "... with ...".
1. A suggestion. You might want to transpose the plots in Figure 2 so that the text is easier to read.


**Strengths And Weaknesses:**

Strengths:
1. The authors presented many evaluation methods (forward probing, reverse probing, a downstream task) over several European languages.
1. The problem studied is an interesting one and I think the method used is also interesting.

Weaknesses:
1. The presentation of crucial concepts lacks clarity. For example, the problem formulation, what does "meaning-bearing units" mean? In the same section, the authors used morphemes as an example but they discourage such comparison elsewhere (in related works section). My first impression is that such *units* would have persistent meanings (like a morpheme) over different inputs but they are not: the slots are computed by a learned neural network for each input (with a learned prior). As stated elsewhere in the paper, each slots computed over an input carries context information (unlike a morpheme).
1. Despite having many summarizing evaluations, it lacks sufficient granular analysis of the learned slots. Such analysis seems necessary when the slots as proposed represent a novel way to encode the meaning of a piece of text.

---

> ### Author Response · Authors · 2023-08-15
> **Response to e62S:**
>
> We would like to thank you for your time reviewing our paper and providing valuable feedback to us.
>
> >The presentation of crucial concepts lacks clarity, Changes 1.
>
> We clarified the problem formulation (Section 2.1) to make it explicit that we want to learn a set-of-vector representation of a text.  This makes the task clearly different from morphology induction or string segmentation.  In particular, we do not expect the model to learn a fixed vocabulary of morphemes, instead computing each vector dynamically from the full text.  However, we still leverage the definition of morpheme as “the smallest meaning-bearing unit of text” to describe what we want these vectors to represent.  Since we are starting from characters, these meaning-bearing units are our target level of granularity for our induced representations.
>
> >Does the ordering of the slots matter (for decoding)?
>
> No, the slots are an unordered set of vectors, but we believe these vectors include positional information as the character representations they are attending to include their positional information.
>
> >Lack of sufficient granular analysis of the learned slot, Changes 2
>
> We would like to note that we have employed two well-established analysis methods in our evaluations, namely visualizing attention maps and probing to show what the slots have learned. It is not clear to us what the reviewer is exactly demanding here, but we have added section 4.3.1 to illustrate the correlation between average slot vectors and positional information. We find that the slots are highly correlated with positional information and probably this information is learned by \mu_i s. This would distribute the slots at different positions at initialization and then, based on the sentence content, the slots gather the information from certain characters around their initial position. The slots would still have the flexibility to determine their boundaries and therefore, represent different units. As we are learning the units dynamically with no morpheme dictionary involved, there is no persistent ‘-ing’ slot to be analysed.
>
> >Would evaluating on an artificial language help clarify what learned slots represent?, Changes 3
>
> This is a very interesting suggestion but beyond the scope of this work. This paper addresses the real patterns of regularities found in real text, and how well the proposed model can capture these patterns.  We try to make as few assumptions as possible about the nature of the real patterns of regularities.   We will consider evaluating on artificial data as future work to better understand Slot Attention, after it has been accepted that Slot Attention is effective on real language data.
>
> 4, 5- Thanks. Well noted.

---

### Author Response · Authors · 2023-08-15
**General Response**

We would like to thank all the reviewers for their time and effort in reviewing our paper and providing constructive feedback. We will address some of the common concerns among the reviewers here.

**Downstream task evaluation**

We are happy that the reviewers found our choice of downstream task as a valid evaluation of the proposed model’s usefulness, but three of the reviewers (21YV, dFr6, JWZR) thought we should have included a comparison to standard segmentation methods, in addition to the stride-based baseline. We have now included a BPE-based model (following reviewer 21YZ’s suggestion) in the results for our ArXiv classification task (section 4.5).  We trained an auto-encoder model in which the inputs and targets are the BPE tokens, and then evaluated the obtained representations in the classification task.  As the results show (Tabel 3), our slot-attention based model outperforms all the baselines, and the BPE-based model falls behind even the character-based model (stride=1). This new baseline adds support to our claims by showing that this downstream task is sensitive to the choice of units, and that the stride-based model is already a strong baseline compared to standard practice.

**Related work**

We thank the reviewers for their suggestions regarding related work (dFr6, JWZR).  This has now been updated accordingly.

---

### Decision · Action_Editors · 2023-09-18

**Recommendation:** Accept as is

**Comment:**

The paper received unanimous "Leaning Accept" recommendations from all four reviewers. They found the main idea of Dynamic Capacity Slot Attention novel and interesting, and the experiments were convincing enough to justify the main claims. The authors have already addressed the major concerns raised by the reviewers, including providing BPE-based baseline results, clarifying the problem formulation, and discussing other related work. Overall, this is a solid submission, and we are pleased to recommend its acceptance.

**Audience:**

This paper should be of interest to the wider audience of TMLR.

**Claims And Evidence:**

The paper extends slot attention, typically found in the computer vision domain, to the text domain. The reviewers appreciated the idea and found it to be novel and interesting. There were initial concerns about insufficient experiments, but the authors addressed them adequately, including the newly reported BPE-based baseline. The reviewers also requested several changes, and the authors properly addressed them in the paper.